# The spatiotemporal association of non-prescription retail sales with cases during the 2009 influenza pandemic in Great Britain

Stacy Todd,[1,2] Peter J Diggle,[3,4] Peter J White,[5,6,7] Andrew Fearne,[8] Jonathan M Read[3]

▶ Prepublication history and additional material is available. To view please visit the journal (http://dx.doi.org/10.1136/bmjopen-2014-004869).

For numbered affiliations see end of article.

**Correspondence to**
Dr Jonathan M Read;
jonread@liv.ac.uk

## ABSTRACT

**Objective:** To assess whether retail sales of non-prescription products can be used for syndromic surveillance and whether it can detect influenza activity at different spatial scales. A secondary objective was to assess whether changes in purchasing behaviour were related to public health advice or levels of media or public interest.

**Setting:** The UK.

**Participants:** National and regional influenza case estimates and retail sales from a major British supermarket.

**Outcome measures:** Weekly, seasonally adjusted sales of over-the-counter symptom remedies and non-pharmaceutical products; recommended as part of the advice offered by public health agencies; were compared with weekly influenza case estimates. Comparisons were made at national and regional spatial resolutions. We also compared sales to national measures of contemporaneous media output and public interest (Internet search volume) related to the pandemic.

**Results:** At a national scale there was no significant correlation between retail sales of symptom remedies and cases for the whole pandemic period in 2009. At the regional scale, a minority of regions showed statistically significant positive correlations between cases and sales of adult 'cold and flu' remedies and cough remedies (3.2%, 5/156, 3.8%, 6/156), but a greater number of regions showed a significant positive correlation between cases and symptomatic remedies for children (35.6%, 55/156). Significant positive correlations between cases and sales of thermometers and antiviral hand gels/wash were seen at both spatial scales (Cor 0.477 (95% CI 0.171 to 0.699); 0.711 (95% CI 0.495 to 0.844)). We found no significant association between retail sales and media reporting or Internet search volume.

**Conclusions:** This study provides evidence that the British public responded appropriately to health messaging about hygiene. Non-prescription retail sales at a national level are not useful for the detection of cases. However, at finer spatial scales, in particular age-groups, retail sales may help augment existing surveillance and merit further study.

## Strengths and limitations of this study

- This study is the first to examine associations between non-prescription retail sales and influenza cases at different spatial resolutions in a British setting and in particular it's potential as part of syndromic surveillance systems.
- The adjustment for seasonality in retail sales was fitted for each spatial resolution in an attempt to capture regional differences which may exist.
- The inclusion of non-pharmaceutical products allowed for the first objective assessment of the response to government public health messaging.
- The main weakness of this study is that regional data were available only for England and for a portion of the 2009/2010 pandemic period.
- Increasing the years of sales data prior to the pandemic period would have provided a more robust estimate of sales trend in a typical year.

## INTRODUCTION

Public health surveillance has traditionally relied on healthcare providers reporting selected notifiable conditions, usually with biological confirmation.[1] Although a key part of national and international health regulations, this system has well-recognised problems including delays in reporting and difficulty in identifying unusual activity.[2] Expansion of non-traditional surveillance methods has occurred over the past two decades, initially because of concerns regarding bioterrorism, and has now been adopted into routine public health systems in many countries. These methods (often referred to as Syndromic Surveillance Systems) offer a real-time or near-real-time collection of data from a variety of sources, ideally in an automated manner which allows early identification of the spread and impact of emerging public health threats and better estimates of incidence in seasonal outbreaks.[3] The 2009

influenza pandemic provided the motivation to adopt and appraise many of these methods.[4][5] In the UK many of the lessons learned during this time were subsequently adopted during the 2012 Olympics and Paralympics to identify any early infectious disease threat.[6]

The surveillance of infectious diseases can be strongly affected by the care-seeking behaviour of individuals.[7] As many individuals will self-medicate for mild illness, surveillance of non-prescription sales has been suggested as an adjunct to healthcare-based surveillance to estimate the magnitude and dynamics of care seeking behaviour.[8] Its usefulness for surveillance of seasonal influenza[9–13] and other illnesses[14–17] has been examined for over 30 years with varying degrees of success. A major potential benefit of this type of surveillance system would be to provide more reliable estimates of incidence when the propensity to seek care is low or changeable, and to identify early-stage epidemics through unusual purchasing activity. Additionally, this type of surveillance may also provide more finely resolved spatiotemporal information on incidence. At present, retail sales are not used for syndromic surveillance in Great Britain.

The first two cases of influenza A H1N1 2009/pdm in the UK were confirmed on 27 April 2009.[18] There was a considerable media response before this and through the summer months. In addition to this a major government campaign was launched ('Catch it, Kill it, Bin it'). This encouraged the use of clean tissues and regular hand washing/use of alcohol hand gel. A leaflet was distributed to every household in the UK on 5 May 2009 with this hygiene advice and also included information on accessing clinical advice.[19] As part of the response within England the National Pandemic Flu Service (NPFS) was established which provided online and telephone advice to individuals including access to antiviral medication, this started on 23 July 2009 and operated until 10 February 2010. This was offered as an alternative to usual primary care services.[20]

We describe the temporal and spatial patterns of sales of over-the-counter flu and cold remedies and non-pharmaceutical products, recommended as part of the advice offered by public health agencies, sold by a major British supermarket. We compare these patterns to national, regional and subregional estimated cases of pandemic influenza during 2009 in Great Britain. We also compare the pattern of sales to national measures of media output and public interest (internet search volume) related to the pandemic during the same time period to assess their relationship to purchasing behaviour.

## METHODS
### Data sources
The weekly estimates of influenza cases were obtained via the Health Protection Agency (HPA; now part of Public Health England) as part of their influenza

surveillance systems (table 1).[21] UK-wide data were calculated via the FluSurvey project (http://www.flusurvey.org) which adjusted healthcare-based surveillance system outputs to account for changes in care-seeking behaviour during the pandemic; the study directly estimated the propensity of individuals to seek care (and therefore contribute to surveillance estimates) during the pandemic through an online survey of a community cohort and indirectly through NPFS consultation.[22] Regional case data were available through the HPA/Q-Surveillance network which monitors diagnoses of influenza-like-illness (ILI) recorded by general practitioners onto routine electronic systems and extracted on a daily and weekly basis.[23] Over 3400 practices contribute to the system, which covers approximately 38% of the UK population; most of the practices are in England with fewer in Wales and Northern Ireland (NI). At the time of the 2009 pandemic no Scottish practices contributed to the system. The density of coverage allows reporting at country and regional levels. Regionally this corresponds to 10 English Strategic Health Authorities (SHAs) and 156 Primary Care Trusts (PCTs), which is the lowest unit of healthcare provision in England with an average population size of 350 000. The HPA/Q-Surveillance data was provided as daily counts of reported ILI cases in each PCT and estimated population in each PCT for that day. This was aggregated to a weekly scale and converted to incidence as rate of cases per 100 000 population. HPA/QSurveillance data were aggregated to three spatial resolutions; subregional, regional and country level (corresponding to PCT, SHA and England, Wales and NI, respectively).

Two measures of media interest and one of public interest over time were compiled (table 1). Daily national newspaper article counts were compiled from the Lexis Nexis newspaper archive,[24] counting articles with headlines containing 'swine flu' or 'h1n1'. The same search phrases were used to identify relevant articles on the Meltwater online database: this database includes newspaper, online, television and radio news articles and reporting.[25] Internet search trends were used as a proxy for public interest in the pandemic. This was derived from Google Insight search facility,[26] and the daily relative volume of searches made where the search terms contained the terms 'swine flu' or 'h1n1' were collated.

Weekly unit sales of non-prescription retail products for a major national UK retailer were obtained for the period 28 January 2008 to 25 April 2010 (table 1). These sales records were derived from a 10% sample of transactions where a loyalty card was presented at the point of purchase and were available at store level. Data on individual product sales were extracted from a master database and aggregated into six categories: Adult Cold and Flu Remedies, Children's Cold and Flu Remedies, Cough Remedies, Thermometers, Anti-Viral Products (including hand gel and wipes), Tissues. Sales as a proportion of customer base were used instead of absolute

sales to control for confounders such as changes in store hours in the period of the study or variation in market share between stores. Short shelf-life products were assumed to be indicative of total customer base. Sales were therefore adjusted in the first instance by dividing weekly total sales (for each category of product and spatial scale) by the average weekly sales of milk and bananas at the appropriate spatial scale (annual sales for 2008 and 2009 available).

The extreme seasonality associated with influenza (and subsequently symptomatic remedies) in temperate zones could introduce biases in the analysis. To adjust for this, an underlying seasonal trend in proportional sales was fitted to log-transformed retail sales data from the beginning of February 2008 to the end of January 2009. This was a prepandemic year, which we assumed to be typical of the seasonal trend in influenza incidence. A flexible way to represent a seasonal trend is through a sum of sine–cosine waves with frequencies corresponding to 1, 2, 3, etc cycles per year. For example, the model with two sine–cosine pairs is

$$\ln (y_t + 1) = \alpha_0 + \alpha_1 \sin \left( \frac{2\pi t}{52} \right) + \beta_1 \cos \left( \frac{2\pi t}{52} \right)$$
$$+ \alpha_2 \sin \left( \frac{4\pi t}{52} \right) + \beta_2 \cos \left( \frac{4\pi t}{52} \right) + \varepsilon_t$$

where $y_t$ is the retail sales data for each week of the year, t, during 2008, $\alpha$ and $\beta$ terms are the regression coefficients for each sine and cosine function, and $\varepsilon$ is an error term.

The model-fitting process was repeated for each product category at each spatial resolution. This resulted in between one and four sine–cosine pairs across the different product groups. In each case, the fitted seasonal model was used to derive weekly residuals for each week of the 2009 and 2010 data; these residuals, which are normalised with respect to normal non-pandemic seasonal sales, are used in the comparative analysis (see online supplementary appendix table A1 and figure A1, A2).

Pearson's correlation was performed between each product category, national UK cases and media reporting. Analysis was performed for the whole pandemic period as well as the early pandemic period (6 April–1 June 2009, media reporting only), summer pandemic wave (1 June–30 August 2009, case and media reporting) and winter pandemic wave (31 August 2009–14 February 2010, case and media reporting). HPA/Q-Surveillance cases were examined at different geographic scales and evaluated by Pearson's correlation coefficients. For each product category, correlation between residual sales and cases was assessed for the period 4 May–9 November 2009. As a rise in retail sales might be expected to occur before an outbreak is detected through healthcare-based surveillance cross correlation with weekly time lags was also performed.

Spatial correlation was performed to look for evidence of clustering of residual sales and influenza cases at different time points. This was performed using the 'spatial test' function in R statistical language, included in the GeoR package[27]: this calculates a test statistic by Monte Carlo permutation testing for spatial autocorrelation based on the use of variograms. For each product group, this test statistic was calculated for subregional residual sales. These spatial correlations were then examined as part of the weekly time series.

All data adjustment and analysis were performed using R statistical software, V.2.15.2. Statistical significance was set at 95%.

## RESULTS

During the declared pandemic period there were two peaks of estimated cases in the summer and winter seasons seen in national flusurvey data (figure 1). HPA/QSurveillance data at a national scale did not show a winter peak. This is most likely due to the established presence of the NPFS service which triaged ILI resulting in a reduced number of primary care consultations. Media reporting was high in the early pandemic period (where there were relatively few cases in the UK) and during the summer wave but was less during the winter wave. Unadjusted national retail sales are shown in figure 1 on a logarithmic scale.

There was a statistically significant positive correlation between thermometer and antiviral product sales and national cases for the whole pandemic period (table 2). When divided into summer and winter pandemic waves, the correlation was stronger in the summer wave than the winter wave. Children's cold and flu remedies were also positively correlated with national cases during the summer wave but not in the winter wave. Correlation between weekly residual sales and weekly media reporting was also performed (see table 2 and online supplementary table A2). Thermometer and antiviral products were significantly positively correlated with media reporting for the whole pandemic period (Cor 0.477 (95% CI 0.171 to 0.699); 0.711 (95% CI 0.495 to 0.844), respectively). No product group sales were significantly associated with media reporting in the early pandemic period though the strength of correlation was higher in the summer than the winter wave (see table 2 and online supplementary table A2).

At a regional level there was no significant correlation between estimated influenza cases and retail sales of adult 'cold and flu' remedies, cough remedies or tissues. There were weak but statistically significant correlations between sales of children's remedies and cases in six English regions and Wales (see online supplementary table A3). Stronger positive correlations were seen between thermometer and cases and hand-gel sales and cases across all English regions and Wales (see online supplementary table A3). No additional significant correlations were identified through cross-correlation analysis. The strongest correlation in cross-correlation testing was for no lag (0 weeks) for all comparisons.

**Table 1** Data sources of influenza case estimates, media reporting and public interest

| Data | Description | Source | Dates | Reference |
|---|---|---|---|---|
| Flusurvey | UK National Case Estimates | Adjusted healthcare-based surveillance system | 1 June 2009–8 February 2010 | 22 |
| HPA/Q-Surveillance network | Regional Case Estimates | General practitioner symptomatic surveillance | 4 May–15 November 2009 | 23 |
| LexisNexis | UK Media Coverage | UK newspaper headlines with reference to A/H1N1pdm and related terms | 25 April–27 December 2009 | 24 |
| Meltwater | UK Media Coverage | UK newspaper headlines, radio and television news items with reference to A/H1N1pdm and related terms | 6 April 2009–19 April 2010 | 25 |
| Google Trends | UK Internet Searches | Internet searches from UK IP addresses with reference to A/H1N1pdm and related terms | 6 April–28 December 2009 | 26 |
| HPA, Health Protection Agency. | | | | |

At the subregional level there was a significant positive correlation between thermometer and hand-gel sales and cases in England (69.9%, 109/156; 71.8%, 112/156, respectively; figure 2). Several subregions had a statistically significant positive correlation between cases and sales of adult 'cold and flu' remedies (3.2%, 5/156) and cough remedies (3.8%, 6/156); however, a greater number of subregions had a significant correlation between cases and children's remedy sales (35.6%, 55/156).

We found periods of significant spatial structure throughout the pandemic period for all sale products (see online supplementary figure A3), particularly for tissue and antiviral product sales which appear to have more sustained periods of spatial patterning than the other product types.

**Figure 1** Top panel: weekly estimated cases of influenza shown are from English general practitioner surveillance system (Health Protection Agency Q–Surveillance) and UK wide estimates adjusted for changes in care seeking behaviour (Flu survey). Middle panel: weekly sales per 100 000 customers of six product groups from a national UK retailer. Bottom panel: scaled weekly estimates of UK media interest (number of relevant newspaper headlines (LexisNexis) or newspaper, radio and television articles (Meltwater)); UK public interest is represented by relative internet search volume from Google Search Trends.

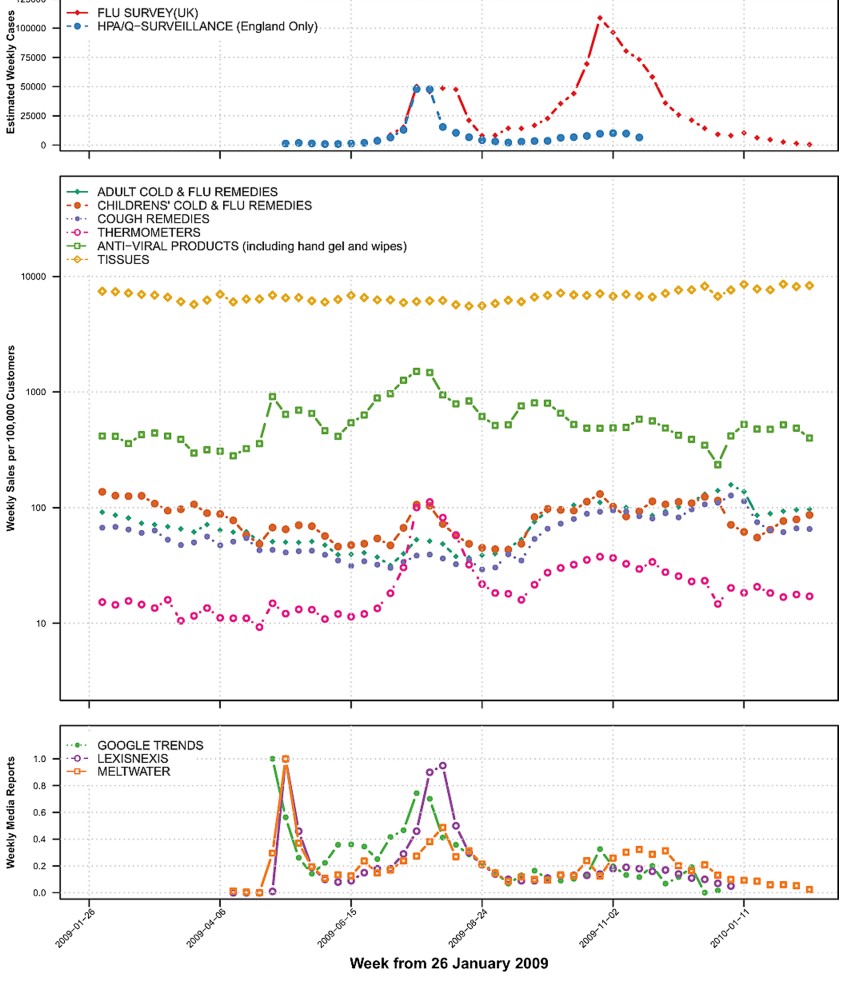

**Table 2** Correlation between retail sales, national cases and media interest

| | Whole pandemic period (19 April 2009–14 February 2010) | | Early pandemic (19 April–31 May 2009) | | Summer wave (01 June–30 August 2009) | | Winter wave (31 August 2009–14 February 2010) | |
|---|---|---|---|---|---|---|---|---|
| | Cor | 95% CI | Cor | 95% CI | Cor | 95% CI | Cor | 95% CI |
| *FluSurvey Case Estimates* | | | | | | | | |
| Adult cold and flu remedies | 0.116 | −0.216 to 0.424 | – | – | 0.193 | −0.401 to 0.672 | 0.149 | −0.270 to 0.521 |
| Childrens' cold and flu remedies | −0.023 | −0.344 to 0.303 | – | – | 0.778** | 0.396 to 0.930 | 0.010 | −0.395 to 0.412 |
| Cough remedies | 0.374* | 0.056 to 0.622 | – | – | 0.245 | −0.353 to 0.702 | 0.396 | −0.009 to 0.689 |
| Thermometers | 0.445** | 0.142 to 0.672 | – | – | 0.935*** | 0.792 to 0.981 | 0.796*** | 0.579 to 0.908 |
| Antiviral products | 0.072 | −0.258 to 0.387 | – | – | 0.671* | 0.190 to 0.892 | 0.014 | −0.392 to 0.415 |
| Tissues | 0.051 | −0.278 to 0.369 | – | – | 0.128 | −0.455 to 0.634 | −0.057 | −0.450 to 0.354 |
| *Meltwater Reports* | | | | | | | | |
| Adult cold and flu remedies | −0.256 | −0.488 to 0.010 | −0.379 | −0.855 to 0.444 | 0.151 | −0.436 to 0.648 | −0.399 | −0.691 to 0.005 |
| Childrens' cold and flu remedies | 0.171 | −0.099 to 0.417 | 0.447 | −0.376 to 0.876 | 0.576* | 0.037 to 0.856 | −0.427* | −0.708 to −0.029 |
| Cough remedies | −0.225 | −0.462 to 0.043 | −0.447 | −0.876 to 0.376 | 0.249 | −0.350 to 0.703 | −0.129 | −0.506 to 0.290 |
| Thermometers | 0.364** | 0.110 to 0.574 | 0.374 | −0.449 to 0.854 | 0.772** | 0.384 to 0.928 | 0.378 | −0.030 to 0.678 |
| Antiviral products | 0.458*** | 0.219 to 0.645 | 0.537 | −0.270 to 0.901 | 0.516 | −0.049 to 0.831 | −0.119 | −0.498 to 0.299 |
| Tissues | −0.288 | −0.514 to −0.025 | 0.386 | −0.437 to 0.858 | 0.241 | −0.358 to 0.699 | −0.451 | −0.723 to −0.059 |
| *Google Searches* | | | | | | | | |
| Adult cold and flu remedies | 0.051 | −0.269 to 0.360 | −0.241 | −0.808 to 0.559 | 0.258 | −0.341 to 0.708 | −0.214 | −0.619 to 0.281 |
| Childrens' cold and flu remedies | 0.369* | 0.060 to 0.613 | 0.452 | −0.371 to 0.877 | 0.716** | 0.273 to 0.909 | −0.303 | −0.674 to 0.191 |
| Cough remedies | −0.050 | −0.360 to 0.270 | −0.318 | −0.836 to 0.498 | 0.295 | −0.306 to 0.728 | −0.083 | −0.529 to 0.399 |
| Thermometers | 0.661*** | 0.437 to 0.808 | 0.212 | −0.579 to 0.797 | 0.891*** | 0.669 to 0.967 | 0.570* | 0.140 to 0.819 |
| Anti-viral products | 0.562*** | 0.299 to 0.745 | 0.346 | −0.474 to 0.845 | 0.610* | 0.089 to 0.869 | 0.038 | −0.437 to 0.496 |
| Tissues | −0.063 | −0.371 to 0.257 | 0.196 | −0.590 to 0.791 | 0.296 | −0.305 to 0.728 | −0.034 | −0.493 to 0.440 |

*<0.05; **<0.01 ***<0.001.

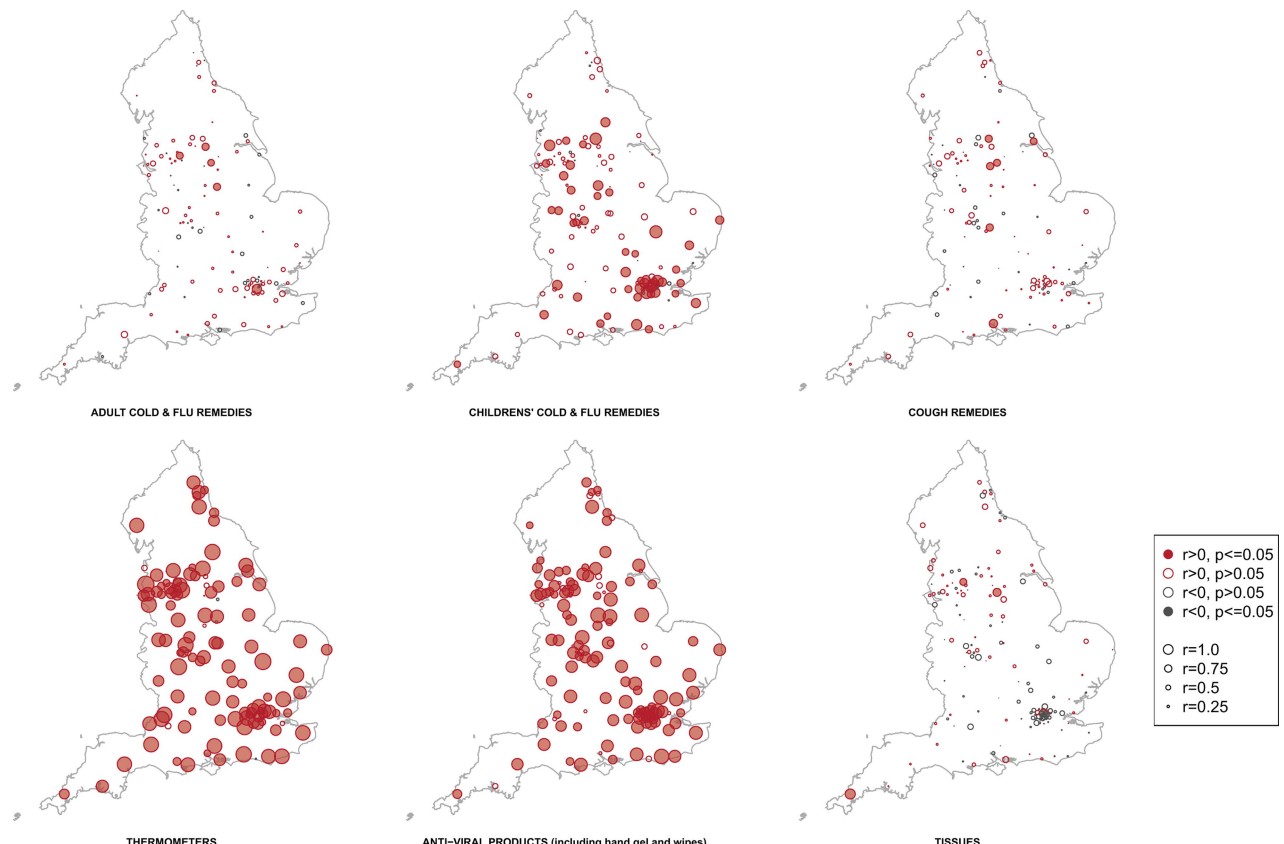

**Figure 2** Correlations between sales of six product categories and influenza A H1N1/pdm cases during 2009. Points relate to a geographic region, size of the point and depth of colour is related to the strength of the correlation.

## DISCUSSION

We analysed non-prescription retail sales data for a major UK supermarket in comparison with pandemic influenza syndromic case estimates within Great Britain to assess the utility of purchase data to reflect case estimates from existing surveillance methods. We found a poor match between symptomatic remedies and cases at the national scale for both summer and winter waves of the pandemic. However, we found a significant association between children's remedies and cases for the summer wave at the national scale, and subregional scales, where we found significant association in 55 of 156 PCTs. Significant positive correlations between cases and sales of thermometers and anti-viral hand gels and hand wash were seen at all spatial scales.

One concern about the use of retail sales as a surveillance tool is that it may be more easily influenced by factors other than symptomatic cases, such as heightened media coverage, and promotional activity by manufacturers, supermarkets and government, than other forms of reporting. The greatest press coverage occurred during the early pandemic period where there were relatively few cases of pandemic influenza in the UK. The lack of correlation between sales and this heightened coverage during this period suggests that 'panic buying' of symptomatic remedies or non-pharmacological groups in response to media reports did not occur. The lack of correlation between sales and media reports in the winter suggests that sales were more driven by cases than media reports as there was a similar level of cases in the summer and winter periods.

The use of sales information for adult and child remedy products has been suggested as a useful augmentation to traditional surveillance mechanisms,[8 13 14] but has not been tested within the UK. Previous studies have suggested that localised retail sales data is more reflective of surveyed influenza patterns than national level data.[11 12 28] Our results broadly support these observations. Some products may be more useful than others in their relative ability to reflect underlying disease incidence.[29] Our results suggest that children's remedies may better reflect community infection patterns than adult products. This may be due to children being at higher risk of infection with 2009 pandemic influenza than adults,[30] being more likely to be symptomatic,[31 32] or may reflect adult–parent differences in self-medication practices.[33] We find no evidence that retails sales may detect cases earlier than established surveillance systems, though our analysis is limited by data resolved at a weekly scale.

Sales of antiviral products and thermometers were highly associated with both pandemic influenza cases and media and public interest measures, especially during the first 'summer' wave of the pandemic. This

finding was not replicated in tissue sales and may reflect larger unit sales per 100 000 customers making signals harder to detect. The use of antiviral products and thermometers (for self-diagnosis) were recommended by UK government public health messaging during the early months of the pandemic and throughout the pandemic.[19] Cross-sectional telephone surveys have generally reported low level of uptake of public health advice[34 35] but there is some evidence that this is a poor indicator of actual behaviour when more objective measures are used.[36] We believe our results are the first national-scale evidence that the public actively responded to these messages, at least through the purchasing of such products, and provides an alternative objective measure of public response to health advice.

There are several limitations to this study. The sales data used here are derived from the shopping purchases of a sample of shopping baskets, and only from purchases involving presentation of a loyalty card. The sales data are only sourced from one supermarket chain, and while that chain has one of the largest market shares nationally in the UK, many non-prescription purchases are likely to be made in other outlets (such as dedicated pharmacies) which may better reflect community incidence of infection. The available sales data, while resolved to purchases made at an individual store level, was only available at a weekly time resolution preventing more finely resolved temporal analysis. Sales of antipyretic medication not branded as 'cold and flu remedies' were excluded from our analysis because of concerns regarding the interpretation of signals from these products. Remedy products may be purchased for a variety of reasons other than to directly medicate against infection with influenza: they could be used for symptom alleviation for a range of other pathogen infections and conditions. We do not know if and how purchasing patterns reflect the use of the products themselves: individuals may use previously purchased products at the onset of new symptoms, only purchasing products when these expire, rather than buying new products to treat a new illness. We did not have access to surveillance data at PCT level for the full pandemic period, which would have been very valuable. The case data to which we compared the retail sales information is largely based on diagnosis of ILI cases (syndromic illness) and not virologically confirmed cases. Case data used in this analysis were not stratified by age; we were therefore unable to perform a more appropriate comparison of case data with adult and children products. Purchasing patterns made over a greater number of years and influenza seasons could have improved the seasonality estimation of purchasing behaviour.

The pandemic of 2009 was of a mild strain, which did not appear to generate a large volume of community cases which self-medicated using OTC remedies and which did not present to existing surveillance mechanisms. However, at particular spatial scales and in particular age-groups, or (we suggest) for more severe strains, retail sales may help augment existing surveillance mechanisms to provide a quantitative indication of care-seeking behaviour. However, there remain considerable uncertainties in the specific usage and self-medicating behaviour of individuals in relation to infection and purchasing of products: further investigation is required prior to the use of sales data for surveillance purposes.

## CONCLUSIONS

Retail sales of over-the-counter symptom remedies at a national level are unlikely to be useful for the detection of cases. However, at more finely resolved spatial scales and in particular age-groups retail sales may help augment existing influenza surveillance and merit further study. Our study demonstrates that the retail sales patterns of particular product types, such as personal hygiene and self-diagnosis products, can be of value in assessing public responses to regional and national health messaging.

**Author affiliations**
[1]Department of Clinical Sciences, Liverpool School of Tropical Medicine, Liverpool, UK
[2]Oxford University Clinical Research Unit, Wellcome Trust Major Overseas Programme, Ho Chi Minh City, Vietnam
[3]Institute of Infection and Global Health, University of Liverpool, Liverpool, UK
[4]Lancaster Medical School, University of Lancaster, Lancaster, UK
[5]Department of Infectious Disease Epidemiology, MRC Centre for Outbreak Analysis and Modelling, Imperial College School of Public Health, London, UK
[6]NIHR Health Protection Research Unit in Modelling Methodology, Department of Infectious Disease Epidemiology, Imperial College School of Public Health, London, UK
[7]Modelling and Economics Unit, Centre for Infectious Disease Surveillance and Control, Public Health England, London, UK
[8]Kent Business School, University of Kent, Canterbury, UK

**Acknowledgements** The authors are grateful to John Edmunds and Ellen Brooks Pollock for providing Flu Survey case estimates and to James Rubin and Susan Michie for providing aggregated media article counts. JMR would like to thank Ashleigh Jellicoe and Xu-Sheng Zhang for assistance in compiling newspaper and retail sales data.

**Contributors** ST performed the analysis and took the lead in writing; JMR conceived the study; all authors designed the analysis and commented on manuscript drafts.

**Funding** ST acknowledges funding from the Wellcome Trust (097465/B/11/Z). JMR acknowledges support for this work from the Economic and Social Research Council (grant ES/K004255/1) and the Medical Research Council (grant RES-355-25-0019). PJW thanks the MRC for Centre funding, and also thanks the UK National Institute for Health Research Health Protection Research Unit (NIHR HPRU) in Modelling Methodology at Imperial College London in partnership with Public Health England (PHE) for funding.

**Competing interests** None. The views expressed are those of the authors and not necessarily those of the NHS, the NIHR, the Department of Health or Public Health England.

**Provenance and peer review** Not commissioned; externally peer reviewed.

**Data sharing statement** Further information on PHE Influenza Surveillance results at http://www.hpa.org.uk

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
