## [Reviewer comments · BMJ Open]

Some articles will have been accepted based in part or entirely on reviews undertaken for other BMJ Group journals. These will be reproduced where possible.

ARTICLE DETAILS

TITLE (PROVISIONAL)	The spatio-temporal association of non-prescription retail sales with cases during the 2009 influenza pandemic in Great Britain.
AUTHORS	Todd, Stacy; Diggle, Peter; White, Peter; Fearne, Andrew; Read, Jonathan

VERSION 1 - REVIEW

REVIEWER	Dr. Avinash Patwardhan Walgreen Co., USA
REVIEW RETURNED	05-Mar-2014

GENERAL COMMENTS	The paper is methodologically sound. Presentation might be improved. I have provided 12 comments in the body of the pdf attached to that effect. I do not have much understanding of how British health care system works but in the USA OTC are used only next to ED visits & outpatient visits for syndromic surveillance by public health agencies (Buehler JW et. al. 2008 in ISDS). I like the methodology of your study but would love to see more studies along the lines. Our (myself & Dr. Bilkovski) paper examined the use of pharmacy script lift; however in the US these data are proprietary in nature & hence not available to scholars or public health agencies. I assume that is not the case in Britain. If so applying the same method for prescription data might be interesting to examine.
--

- The reviewer also provided a marked copy with comments. Please contact the publisher for full information.

REVIEWER	Tom Fowler and Carol Chatt Field Epidemiology Service Public Health England UK
REVIEW RETURNED	27-Mar-2014

GENERAL COMMENTS	This is a very interesting paper on a novel topic, which despite negative findings is of national and international interest. We believe the findings may be of particular interest to other countries where the healthcare infrastructure limits co-ordinated approaches to syndromic surveillance. There are however a number of issues that need to be addressed. Major issues: 1. A comparison is made to a syndromic surveillance system currently used in the UK. However, there is no exploration around
--

	whether any of the different data sources considered can add value to or replace the current syndromic surveillance systems in place, e.g. ability to detect smaller signals, earlier signals etc. This needs to be addressed in the introduction and the discussion. 2. There is a lack of clarity throughout the paper regarding the output of syndromic surveillance systems. Unlike traditional lab-based surveillance systems, QSurveillance does not identify confirmed cases but rather is set up to identify signals of reports of influenza-like illness. It would be better to describe the outputs in terms of signals regarding influenza activity. 3. The purpose and objectives of the paper need clarifying. Having read the paper, we believe these to be a) assessment of whether or not retail sales can be used for syndromic surveillance, b) assessment of whether the use of retail sales can be used as part of syndromic surveillance systems in place c) whether use of retail sales data can be used to assess the effectiveness of public health advice. This needs to be clarified in the abstract and the introduction. It is further unclear whether the comparison to media and Internet search data is to assess the utility of retail sales in comparison to these methods of surveillance, currently implemented methods of syndromic surveillance or to evaluate public health messages. Minor issues: 1. The paper would benefit from specifically describing the purpose and benefit of comparison of each data source, e.g. why compare Q-Surveillance and media coverage? why compare Q-Surveillance and Internet searches? 2. Mention is made of lagged comparisons which would help identify if retail sales provide earlier detection but this is not subsequently reported in the results. 3. The timing of the start of public health advice is not highlighted therefore it is difficult to assess its effectiveness 4. Regarding figure 1 (top panel) mention is made of 2 peaks in both the Flu Survey (UK) and Q-Surveillance data although there appears to be only 1 peak in the Q-Surveillance data. 5. In the introduction, the rationale is given that retail sales may be useful in the early assessment of novel influenza epidemics as supposed to surveillance of seasonal influenza. No specific discussion of this point occurs in the conclusions. 6. There are some typos e.g. page 4 line 49 'though an online survey', page 4 line 54 'Northern Island', page 5 line 4 'including should be included, page 6 line 47 0.699 should be 0.699), Page 7 line 14 that should be than, Page 7 line 45 the should be deleted. Page 7 line 46 counties should be countries, Page 8 line 38 self-mediated should be self-medicated. Interchangeable use of the terms British and 'the GB'.
--	--

VERSION 1 – AUTHOR RESPONSE

Reviewer Name Dr. Avinash Patwardhan

Institution and Country Walgreen Co., USA

Please state any competing interests or state 'None declared': None declared

The paper is methodologically sound. Presentation might be improved. I have provided 12 comments in the body of the pdf attached to that effect. I do not have much understanding of how British health care system works but in the USA OTC are used only next to ED visits & outpatient visits for syndromic surveillance by public health agencies (Buehler JW et. al. 2008 in ISDS). I like the

methodology of your study but would love to see more studies along the lines. Our (myself & Dr. Bilkovski) paper examined the use of pharmacy script lift; however in the US these data are proprietary in nature & hence not available to scholars or public health agencies. I assume that is not the case in Britain. If so applying the same method for prescription data might be interesting to examine.

We agree with the reviewer that more studies of this type of needed. NHS Prescription data is slowly becoming publicly available but so far is only available aggregated to monthly periods (please see <http://bmjopen.bmj.com/content/3/1/e001363>). We do intend to investigate the use of such data for infectious disease surveillance and modelling purposes in future studies.

2 of 25, 8-9: This is an informative statement about the state of art and might better suit in the introduction since it does not reflect the results of the study. (as already done in 4 of 25: 25,26)

We have removed this statement from the abstract, and retain the statement in the introduction: "Its [surveillance of non-prescription sales] usefulness for surveillance of seasonal influenza [9-13] and other illnesses [14-17] has been examined for over 30 years with varying degrees of success."

2 of 25, 11-12: The language may need more clarity, re. compare during influenza season or in influenza cases...

We have changed the abstract to try and make the overall objectives of the study clearer.

"To assess whether retail sales of non-prescription products can be used for syndromic surveillance and whether it can detect influenza activity at different spatial scales. A secondary objective was to assess whether changes in purchasing behaviour related to public health advice or levels of media or public interest."

2 of 25, 17: There is no reference to recommendations by the agencies. Might be good to add (though you mention a bit about it at 8 of 25, 4-5- pointing to reference 31)

Because of word count limits we have not added more detail regarding the specific recommendations within the abstract. More detail has been added within the main manuscript. See response to comment 8 of 25, 10-11.

2 of 25, 35: Same as comment 3 above.

See above.

2 of 25, 35-36: May be want to say non prescription retail sales or something along that line

We have changed the text to read:

"Non-prescription retail sales at a national level are not useful for the detection of cases."

3 of 35, 7-8: [http://www.northwest-](http://www.northwest-zoonoses.info/writedir/896aJonRead_Novel_Surveillance_Methods.pdf)

[zoonoses.info/writedir/896aJonRead_Novel_Surveillance_Methods.pdf](http://www.northwest-zoonoses.info/writedir/896aJonRead_Novel_Surveillance_Methods.pdf) same authors but published where?

This PowerPoint presentation was preliminary analysis of the submitted work presented at a local workshop meeting. It has not been published in peer reviewed journals previously.

4 of 25, 44: Might help at this stage to provide the period for which estimates were made

We have included the dates within Table 1.

4 of 25, 46, 47: Might need clarification about adjustment

The adjustment uses community based online cohort to estimate dynamic changes in health care seeking behaviour; the paper referenced (Brooks-Pollock et al) contains full details and we direct the reviewer to that paper for further details. However, we have expanded the text slightly to:

“UK-wide data was estimated via the FluSurvey project (www.flusurvey.org) which adjusted healthcare-based surveillance system outputs to account for changes in care-seeking behaviour during the pandemic; the study directly estimated the propensity of individuals to seek care (and therefore contribute to surveillance estimates) during the pandemic through an online survey of a community cohort and indirectly through NPFS consultation [22]”

7 of 25, 46: Well, not exactly so, e.g. the paper: Sugawara T, Ohkusa Y, Shigematsu M, Taniguchi K, Murata A, Okabe N. An experimental study for syndromic surveillance using OTC sales *Kansenshogaku Zasshi*. 2007 May;81(3):235-41. suggest it might have some utility but in general yes. Thank you for pointing out this reference. Our reference 12 (Ohkusa et al *MMWR* 2005) appears to be the English language version of this publication with the same authors, data set and figures. Having reviewed this version we agree with the authors conclusion that it is unlikely to be useful at a national level. We have therefore modified our statement to try and better reflect the published literature: “Previous studies have suggested that localised retail sales data is more reflective of surveyed influenza patterns than national level data.”

8 of 25, 10-11: Might want to explain a bit why public responded to some products and not all if the recommendations were more than the above two

The key public health message in the UK was the “Catch it, Kill it, Bin it” campaign. This encouraged the use of clean tissues and regular hand washing/use of alcohol hand gel. The taking of temperature was encouraged to aid diagnosis. The other response was a telephone based advice line which triaged people to self care (symptomatic management with anti-pyretic/other remedies) or to receive anti-virals (with or without review by healthcare personnel). Detailed analysis of NPFS (National Pandemic Flu Service – responsible for anti-viral distribution) is published elsewhere and has been added to our article. We now include a new section in the introduction describing in more detail the healthcare response, and setting the scene for the reader:

“The first two cases of influenza A H1N1 2009/pdm in the UK were confirmed on 27 April 2009 [18]. There was a considerable media response before this and through the summer months. In addition to this a major government campaign was launched (“Catch it, Kill it, Bin it”). This encouraged the use of clean tissues and regular hand washing/use of alcohol hand gel. A leaflet was distributed to every household in the UK on 5 May 2009 with this hygiene advice and also included information on accessing clinical advice [19]. As part of the response within England a National Pandemic Flu Service (NPFS) was established which provided online and telephone advice to individuals including access to anti-viral medication, this commenced on 23 July 2009 and operated until 10 February 2010. This was offered as an alternative to usual primary care services [20].”

We have modified the discussion regarding the lack of signal in tissue sales and our omission of anti-pyretic not branded as ‘cold and flu remedies’.

“This finding was not replicated in tissue sales and may reflect larger unit sales per 100,000 customers making signals harder to detect.”

“Sales of anti-pyretic medication not branded as ‘cold and flu remedies’ were excluded from our analysis because of concerns regarding the interpretation of signals from these products. Remedy products may be purchased for a variety of reasons other than to directly medicate against infection with influenza: they could be used for symptom alleviation for a range of other pathogen infections and conditions.”

Ref 3, link does not work reference 31 the link did not work
Fixed. Thank you.

Reviewer Name Tom Fowler and Carol Chatt
Institution and Country Field Epidemiology Service
Public Health England
UK

Please state any competing interests or state 'None declared': None declared

This manuscript was reviewed in partnership by Tom Fowler and Carol Chatt.

This is a very interesting paper, which despite negative findings is of national and international interest. We do have some issues with the manuscript currently, which we have clarified in the notes to the authors. We would strongly recommend this paper is published if the authors address the issues raised. This is a very interesting paper on a novel topic, which despite negative findings is of national and international interest. We believe the findings may be of particular interest to other countries where the healthcare infrastructure limits co-ordinated approaches to syndromic surveillance.

There are however a number of issues that need to be addressed.

Major issues:

1. A comparison is made to a syndromic surveillance system currently used in the UK. However, there is no exploration around whether any of the different data sources considered can add value to or replace the current syndromic surveillance systems in place, e.g. ability to detect smaller signals, earlier signals etc. This needs to be addressed in the introduction and the discussion.

Our primary intention with this study (as stated below) was not to assess how retail sales (or other data sources such as internet searches) may be used as additional information for established surveillance systems, but rather whether retail sales patterns reflect best estimates of cases in general. We feel this is a necessary first step prior to considering whether and how such information may be used in an augmentation role. Given our findings we do not expect retail sales to replace existing surveillance mechanisms. However, we have revised our abstract objectives and modified our discussion to be clearer:

“Objective: To assess whether retail sales of non-prescription products can be used for syndromic surveillance and whether it can detect influenza activity at different spatial scales. A secondary objective was to assess whether changes in purchasing behaviour related to public health advice or levels of media or public interest.”

“We analysed non-prescription retail sales data for a major GB supermarket in comparison with cases of pandemic influenza within Great Britain to assess the utility of purchase data to reflect case estimates from existing surveillance methods.”

2. There is a lack of clarity throughout the paper regarding the output of syndromic surveillance systems. Unlike traditional lab-based surveillance systems, QSurveillance does not identify confirmed cases but rather is set up to identify signals of reports of influenza-like illness. It would be better to describe the outputs in terms of signals regarding influenza activity.

We have modified our description of the QSurveillance scheme in our methods section to be clearer:

“Regional case data was available through the HPA/Q-Surveillance network which monitors diagnoses of influenza-like-illness (ILI) recorded by general practitioners onto routine electronic systems and extracted on a daily and weekly basis [21].”

and

“The HPA/Q-Surveillance data was provided as daily counts of reported ILI cases in each PCT and including estimated population in each PCT for that day.”

We also include a sentence in the limitations section of the discussion highlighting the basis of our case data:

“The case data to which we compared the retail sales information is largely based on diagnosis of influenza-like illness cases (syndromic illness) and not virologically confirmed cases.”

3. The purpose and objectives of the paper need clarifying. Having read the paper, we believe these to be a) assessment of whether or not retail sales can be used for syndromic surveillance, b) assessment of whether the use of retail sales can be used as part of syndromic surveillance systems in place c) whether use of retail sales data can be used to assess the effectiveness of public health advice. This needs to be clarified in the abstract and the introduction. It is further unclear whether the comparison to media and Internet search data is to assess the utility of retail sales in comparison to these methods of surveillance, currently implemented methods of syndromic surveillance or to evaluate public health messages.

Our primary aim for this study was assessing whether retail sales can be used for syndromic surveillance. While retail sales data may be more appropriately used as an augmentation of existing surveillance systems, we do not attempt to address this issue. A secondary aim was to attempt to detect signals of purchasing that may indicate responses to public health messages. We have adjusted our stated aims to better reflect our motivation and analysis, firstly in the abstract:

“Objective: To assess whether retail sales of non-prescription products can be used for syndromic surveillance and whether it can detect influenza activity at different spatial scales. A secondary objective was to assess whether changes in purchasing behaviour related to public health advice or levels of media or public interest.”

and in the introduction:

“Here, we describe the temporal and spatial patterns of sales of over-the-counter flu and cold remedies and non-pharmaceutical products (recommended as part of the advice offered by public health agencies) sold by a major British supermarket. We compare these patterns to national, regional and sub-regional estimated cases of pandemic influenza during 2009 in Great Britain. We also compare the pattern of sales to national measures of media output and public interest (internet search volume) related to the pandemic, during the same time period, to assess their relationship to purchasing behaviour.”

Minor issues:

1. The paper would benefit from specifically describing the purpose and benefit of comparison of each data source, e.g. why compare Q-Surveillance and media coverage? why compare Q-Surveillance and Internet searches?

We do discuss in the narrative the difference between i) retail sales and cases associations and ii) retail sales and public or media interest associations. We do not directly compare any case estimates with media volume or internet searches. While we assume media and internet search data to represent media and public interest in the pandemic, and see large changes throughout the

pandemic, our study is focussed on how this may relate to retail sales, not to assess the potential biases in established surveillance systems. That may be a useful study but is beyond the scope of this manuscript.

2. Mention is made of lagged comparisons which would help identify if retail sales provide earlier detection but this is not subsequently reported in the results.

We now state in the results section:

“The strongest correlation in cross-correlation testing was for no lag (0 weeks) for all comparisons.”

and in the discussion section:

“We find no evidence that retail sales may detect cases earlier than established surveillance systems, though our analysis is limited by data resolved at a weekly scale.”

3. The timing of the start of public health advice is not highlighted therefore it is difficult to assess its effectiveness

We now include a more detailed description in the introduction regarding the public health response to the pandemic in GB:

“The first two cases of influenza A H1N1 2009/pdm in the UK were confirmed on 27 April 2009 [18]. There was a considerable media response before this and through the summer months. In addition to this a major government campaign was launched (“Catch it, Kill it, Bin it”). This encouraged the use of clean tissues and regular hand washing/use of alcohol hand gel. A leaflet was distributed to every household in the UK on 5 May 2009 with this hygiene advice and also included information on accessing clinical advice [19]. As part of the response within England a National Pandemic Flu Service (NPFS) was established which provided online and telephone advice to individuals including access to anti-viral medication, this commenced on 23 July 2009 and operated until 10 February 2010. This was offered as an alternative to usual primary care services [20].”

4. Regarding figure 1 (top panel) mention is made of 2 peaks in both the Flu Survey (UK) and Q-Surveillance data although there appears to be only 1 peak in the Q-Surveillance data.

We now state at the start of the results section:

“During the declared pandemic period there were two peaks of estimated cases in the summer and winter seasons seen in national flusurvey data (Figure 1). HPA/QSurveillance data at a national scale did not show a winter peak. This is most likely due to the established presence of the NPFS service which triaged influenza like illness resulting in a reduced number of primary care consultations.”

5. In the introduction, the rationale is given that retail sales may be useful in the early assessment of novel influenza epidemics as supposed to surveillance of seasonal influenza. No specific discussion of this point occurs in the conclusions.

The introduction has been changed to emphasise the role of syndromic surveillance in seasonal outbreaks as well as pandemic spread.

“These methods (often referred to as Syndromic Surveillance Systems) offer a real-time or near-real-time collection of data from a variety of sources, ideally in an automated manner which allows early identification of the spread and impact of emerging public health threats and better estimates of incidence in seasonal outbreaks.”

In the discussion we now discuss our findings regarding cross-correlation analysis that we used to assess the ability of retail sales to detect changes in cases earlier than established systems. "We find no evidence of that retail sales may detect cases earlier than established surveillance systems, though our analysis is limited by data resolved at a weekly scale."

6. There are some typos e.g. page 4 line 49 'though an online survey', page 4 line 54 'Northern Island', page 5 line 4 'including should be included, page 6 line 47 0.699 should be 0.699), Page 7 line 14 that should be than, Page 7 line 45 the should be deleted. Page 7 line 46 counties should be countries, Page 8 line 38 self-mediated should be self-medicated. Interchangeable use of the terms British and 'the GB'.

Thank you. All have now been corrected.